# A Comparative Genomics Approach for Analysis of Complete Mitogenomes of Five Actinidiaceae Plants

**DOI:** 10.3390/genes13101827

**Published:** 2022-10-09

**Authors:** Jun Yang, Chengcheng Ling, Huamin Zhang, Quaid Hussain, Shiheng Lyu, Guohua Zheng, Yongsheng Liu

**Affiliations:** 1College of Horticulture, Anhui Agriculture University, Hefei 350002, China; 2State Key Laboratory of Subtropical Silviculture, Zhejiang A&F University, 666 Wusu Street, Hangzhou 311300, China; 3College of Horticulture, Fujian Agriculture and Forestry University, Fuzhou 350002, China

**Keywords:** Actinidiaceae, mitogenome, comparative analysis, phylogenetic analysis

## Abstract

Actinidiaceae, an economically important plant family, includes the *Actinidia*, *Clematoclethra* and *Saurauia* genus. Kiwifruit, with remarkably high vitamin C content, is an endemic species widely distributed in China with high economic value. Although many Actinidiaceae chloroplast genomes have been reported, few complete mitogenomes of Actinidiaceae have been studied. Here, complete circular mitogenomes of the four kiwifruit species and *Saurauia tristyla* were assembled. Codon usage, sequence repeats, RNA editing, gene transfers, selective pressure, and phylogenetic relationships in the four kiwifruit species and *S**. tristyla* were comparatively analyzed. This research will contribute to the study of phylogenetic relationships within Actiniaceae and molecular barcoding in kiwifruit.

## 1. Introduction

According to the endosymbiosis theory, the mitochondrion is an endosymbiotic alphaproteobacterium engulfed by the archaeal-derived host cell and eventually evolves into a semiautonomous organelle [1,2,3]. Mitochondria, known as energy factories, play a crucial role in numerous metabolic processes related to energy generation, synthesis, and degradation in living cells [4]. Mitochondrial DNA is maternally inherited in most seed plants [5]. With genome sequencing technology’s rapid development, various complete organelle genomes in plants have been extensively studied [6]. Nearly 7576 chloroplasts and mitogenomes of land plants have been published in the National Center for Biotechnology Information (https://www.ncbi.nlm.nih.gov/genome/browse#!/organelles/) (accessed on 10 January 2022). The number of mitogenomes in land plants published was less than 20 before 2015 (Appendix A). In recent years, it is no doubt that the number of land plant mitogenomes has increased significantly from 2018 to 2021 (Appendix A). However, compared to the completed chloroplast genomes (7246), only 324 completed mitogenomes were assembled (Appendix A), suggesting that the interpretation and functional annotation of the mitochondrial genome is complex in comparison to other organelles.

The intergenomic DNA transfers and highly dynamic, multipartite structures of plant mitogenomes may make it challenging to build plant mitogenomes [7]. Several articles have reported that most plant mitogenomes range from 200 to 2000 kb in size [8]. Differences in mitogenome size can be attributed to repetitive sequences and foreign DNA derived from other organisms during evolution [9]. Many intramolecular recombination events and subgenomic conformations have been found in some land plants, such as *Scutellaria tsinyunensis* [10], *Cucumis sativus* [11], *Ipomoea batatas* [12], and *Brassica napus* [13]. In extreme environments, gene loss and RNA edits may occur during plant mitogenome rearrangement [14]. Several separate chromosomes can be found in some higher plant mitochondrial genomes. The cucumber mitogenome, for instance, has three separate chromosomes [11]. The mitogenomes of *Globodera ellingtonae* and *Camellia sinensis* have two separate chromosomes [15,16]. The mitogenomes of the plant’s seeds contain many repeating sequences, including simple sequence repeats (SSRs), tandem repeats, and scattered repeats. In addition, there are also many insertions/deletions (indels) and single nucleotide polymorphisms (SNPs) within mitogenomes [17,18]. SSRs and SNPs have been widely applied to identify species rapidly and for phylogenetic plant analyses, especially in Chinese herbal medicine classification [19,20]. Moreover, it is an essential feature for mitogenome evolution via intracellular transfer between the mitochondria and the chloroplast genomes [21]. Most of the transferred sequences are transferred from the nucleus to the mitochondria, but several chloroplast-derived tRNA genes are transferred to the mitochondria and perform essential functions [22]. The horizontal gene transfer (HGT) phenomenon also plays a significant role in the evolution of plant mitogenomes [9]. These findings suggested the existence of instability in higher plants’ mitogenome structures. Finally, a long-reads strategy in combination with short-reads technologies (Pacbio SMRT, Oxford Nanopore, or Illumina mate-pair) were applied to solve the problem caused by this structural instability in mitogenome assembly.

Actinidiaceae, an economically important plant family, includes the *Actinidia*, *Clematoclethra,* and *Saurauia* genus [23,24]. Among the Actinidiaceae family of the Asterids, kiwifruit with remarkably high vitamin C content, commonly known as ‘the king of fruits’, is an economically important horticultural fruit tree. Kiwifruit is widely cultivated in Asia, Europe and Oceania (https://www.fao.org/faostat/zh/#data/QCL, Appendix A) (accessed on 3 December 2021). Worldwide annual kiwifruit production increased rapidly from 2012 to 2020 and reached approximately 2 million tons in 2020 (https://www.fao.org/faostat/zh/#data/QCL, Appendix A) (accessed on 3 December 2021). Total kiwifruit production in Asia was the highest, accounting for 52.5%, followed by Europe (25.3%) from 2012 to 2020 (https://www.fao.org/faostat/zh/#data/QCL, Appendix A) (accessed on 3 December 2021). It is noted that the annual kiwifruit production in China was the highest in Asia and reached up to 1.49 million tons in 2020 (https://www.fao.org/faostat/zh/#data/QCL, Appendix A) (accessed on 3 December 2021). This may be due to the abundant kiwifruit germplasm resources in China. So far, diploid *A. chinensis* and hexaploid *A. chinensis* var deliciosa are the most commercial kiwifruit varieties. Abiotic and biotic stresses, including drought, salinity, low or high temperatures, and Pseudomonas syringae pv. actinidiae (Psa) seriously affect the yield and quality of kiwifruit [25]. After incidence of Psa, there is no remedy available to control it, except for destroying the tree to prevent the spread of the disease. Thus, Psa seriously threatens the production and development of the kiwifruit industry worldwide. It has been reported that *A. eriantha* var ‘huate’ and *A. chinensis* var deliciosa ‘jinkui’ strongly resist Psa [26,27]. Mitochondria genetic engineering would be beneficial in developing a method of resilience to abiotic and biotic stresses [25]. Hence, the complete mitogenomes of kiwifruit are sequenced, providing great promise for breeding kiwifruit cultivars with resilience to abiotic and biotic stresses.

For the last three decades, 4 kiwifruit nuclear genomes and over 29 complete chloroplast genomes from the Actinidiaceae family have been sequenced [28,29,30,31,32], while no complete mitogenome of this family has been reported previously. To elucidate the evolutionary mechanisms and structural features that underlie the Actinidiaceae family’s mitogenomic diversity, the complete mitochondria genome of the diploid *A. chinensis*, and the tetraploid *A. chinensis*, hexaploid *A. chinensis* var deliciosa, *A. eriantha* and *S. tristyla* were sequenced and assembled in this study. The mitochondria genome with a two-chromosomal conformation was found in diploid *A. chinensis*, tetraploid *A.chinensis* and *S. tristyla*. The genome size (939 kb) of *A. chinensis* var deliciosa was significantly more extensive than other Actiniaceae species. Therefore, we hypothesized that the mitogenome may experience expansion during *A. chinensis* ploidy doubling. We analyzed the mitogenome structures of four kiwifruit species and *S. tristyla* to elucidate/unveil the genomic repeats, RNA editing sites, relative synonymous codon usage, gene transfer, and the evolutionary relationships among the Actinidiaceae family. To sum up, our study will be instrumental for genetic engineering and breeding programs.

## 2. Materials and Methods

### 2.1. Plant Materials and Genome Sequencing

Appendix A shows details of the tested materials. Fresh leaves were wrapped in aluminum foil, flash frozen in liquid nitrogen, and stored at −80 °C for subsequent use. High-quality total genomic DNA was extracted using a DNAsecure Plant Kit (Tiangen Biotech, Co. Ltd., Beijing, China). The DNA library construction and sequencing were performed as previously reported by Emerman et al. [33].

### 2.2. Mitogenome Assembly and Annotation

The Oxford Nanopore long-reads were de novo assembled for the five mitogenomes using SMARTdenovo with default parameters [34]. To obtain high-quality mitogenomes, the Illumina short-reads were conducted after polishing with minimap2/miniasm [35], racon (v1.4.20) [36] and pilon (v1.23) [37] to correct nanopore long-read errors. Furthermore, we used the BWA [38] and SAMtools [39] to map all the raw reads to the assemble mitogenomes. In the last step, the assembled PacBio sequences were checked for overlaps and joined. Mitochondria annotations were achieved using the online Geseq tool [40] with *Actinidia arguta* as the reference mitogenomes from GenBank:MH559343. We manually edited the annotation problems, using Apollo [41], and OGDRAWv1.3.1 [42] to draw the circular maps of the mitogenomes. All transfer RNA genes were checked by the online tRNAscanSE service (http://lowelab.ucsc.edu/tRNAscan-SE/, accessed on 1 January 2022) [43].

### 2.3. Repeat Sequences and Chloroplast to Mitochondrion DNA Transformation

The SSR (simple sequence repeats), including mono-, di-, tri-, tetra-, penta-, and hexanucleotide bases pairs with 12, 6, 4, 3, 3, and 3 repeat numbers, respectively, were detected using the microsatellite identification tool MISA-web55 (https://webblast.ipk-gatersleben.de/misa/, accessed on 3 February 2022) with default parameters [44]. Tandem Repeats Finder v4.09 software [45] (http://tandem.bu.edu/trf/trf.submit.options.html, accessed on 5 February 2022) with default parameters was employed to detect tandem repeats (>6 bp repeat units). The chloroplast fragments’ insertion in the mitogenome was identified using the BLASTN tool according to the following screening criteria: matching rate ≥ 70%, E-value ≤ 1 × 10^−6^, and length ≥ 40 [46]. Circos maps were visualized using the advanced circos module in Tbtools [47].

### 2.4. RNA Editing Predicting and Codon Usage

We used the online PREP-Mt suite of servers (http://prep.unl.edu/, accessed on 5 March 2022) [48], with a cutoff value of 0.2, to predict the RNA editing sites of the 39 protein-coding genes of the 5 mitogenomes. The relative synonymous codon usage (RSCU) was calculated by MEGA X53 [49].

### 2.5. Substitution Rate Calculation and Phylogenetic Inference

Pairwise 19 protein-coding gene sequences of the mitogenomes of Actinidia were used to estimate the pairwise nucleotide substitution rates, including the non-synonymous substitution rate (Ka) and synonymous substitution rate (Ks), and the ratio of Ka to Ks. The Ka/Ks ratios were calculated by PAML (v4.9) [50] using the yn00 module with default parameters. The Ka/Ks values’ heatmap was plotted using Tbtools [47]. In order to further analyze the phylogenetic position of the Actinidiaceae species, 23 plant mitogenomes from GenBank were downloaded for phylogenetic tree construction. A total of 20 orthologous mitochondrial genes were identified and extracted using PhyloSuite (v1.2.1) [51]. The corresponding nucleotide sequences were aligned using MAFFT (v7.450) [52] implemented in PhyloSuite. The phylogenetic tree was constructed using the maximum likelihood (ML) method via RAxML v8.1.5 [53] with 1000 bootstrap replicates. Furthermore, the web iTOL (https://itol.embl.de, accessed on 8 April 2022) [54] was used to visualize the phylogenetic trees.

## 3. Results

### 3.1. Mitogenome Assembly, Annotation and Gene Features

The de novo assembly assembled five complete mitogenomes of Actiniaceae species. The de novo genome assembly yielded a single circular molecule for *A. chinensis* var deliciosa (939 kb) and *A. eriantha* (768 kb). In contrast, two distinct circular chromosomal genomes for *A. chinensis* (2×), *A. chinensis* (4×) and *S. tristyla* (Figure 1A–D and Appendix A) were reported. *A. chinensis* (2×) and *A. chinensis* (4×) mitogenomes exhibited similar genome size (916 kb vs. 907 kb). The genome size (939 kb) of *A. chinensis* var deliciosa was significantly more extensive than the other Actiniaceae species. *S. tristyla* (482 kb) has the smallest genome size among them. Interestingly, their mitogenomes, containing similar GC content, are about 46% (Appendix A). A comparison of the annotated genes in *Actiniaceae* revealed that the pseudogene rps2 is absent in tetraploid and hexaploid *A. chinensis* (Appendix A). The loss of the *sdh4* gene in the *A. arguta* mitochondrion genome was also notable (Appendix A).

### 3.2. Repeat Sequences and Chloroplast-Derived Region Analysis

As shown in Figure 2A, a total of 49–124 tandem repeats were found in the Actiniaceae mitogenome. The number of tandem repeats, comprising between 10 and 20 bp in the *A. chinensis* (2×) mitogenome, was significantly lower for the other species. Compared to the tetraploid and hexaploid *A. chinensis*, the number ranged from 40 bp to 105 bp in diploid *A. chinensis*. About half of the tandem repeats ranged from 10 bp to 20 bp in kiwifruit, whereas most of the tandem repeats ranged from 41 bp to 105 bp in *S. tristyla*. *A. eriantha* contained the fewest tandem repeats. A wealth of SSRs was identified (Figure 2). SSRs in hexamers were discovered in all species except *S. tristyla* (Figure 2C). Nearly 78% of the SSRs belonged to monomers and dimers (Figure 2B). Tetramers and pentamer-nucleotide repeats were less frequent in the Actiniaceae mitogenome. *A. chinensis* mitogenomes contained a higher number of SSRs than those of *S. tristyla* and *A. eriantha*. *S. tristyla* had the lowest number of SSRs (Figure 2B).

Plastid-derived sequences were detected in five *Actiniaceae* species’ mitogenomes. Three species of chloroplasts and mitogenomes have high and widespread homologies in *A. chinensis* (2×–6×) (Figure 3E). A total fragment length of 50–55 kb of the tandem repeats red transferred fragment, which accounts for about 1/3 of the chloroplast genome, was identified in the *A. chinensis* (2×–6×) mitogenome (Figure 3F). Plastid-derived sequences in *A. chinensis* (2×–6×) were significantly higher in number than in *A. eriantha* and *S. tristyla* (Figure 3F). Five intact chloroplast genes (rpoC1, ndhB, rps7, rps19, and rpl23) were transferred into the mitogenome in the *A. chinensis* (2×–6×) species (Appendix A).

### 3.3. RNA Editing Sites and Codon Usage Analysis of PCGs

As shown in Figure 4, the RNA editing sites of 39 PCGs of the mitogenomes of 5 Actinidiaceae plants were predicted in this study. Three cytochrome c biogenesis genes, including ccmFn, ccmB, and ccmC, displayed the most RNA editing sites in five Actinaceae sp. plants. Interestingly, we found that the number of *NAD1* gene RNA editing sites in *A. chinensis* var deliciosa was significantly higher than in the other species (Figure 4). Only the *rpl2* gene in the *A. chinensis* (4×) had no RNA editing sites (Figure 4). The *rpl16*, *rps1*, and *rps2* genes contained the same number of editing sites in the *A. chinensis* (2×) and *A. eriantha*, but not in the other species (Figure 4).

The codon distribution and relative synonymous codon usage (RSCU) of five Actinidiaceae species’ mitogenomes were analyzed. The RSCU analysis showed that Leu, Ser, and Arg appeared the most frequently, whereas those that encoded Met and Trp were relatively less abundant in five Actinidiaceae species’ mitogenomes. Five species in the Actinidiaceae family share a similar RSCU style (Figure 5A–E).

### 3.4. The Synonymous and Nonsynonymous Substitution Rate (Ka/Ks) and Phylogenetic Analysis

Nineteen protein-coding genes of six Actinidiaceae mitogenomes were used to calculate the Ka/Ks ratios. As shown in Figure 6, we observed that the *sdh3* gene had an abnormally high Ka/Ks ratio > 1 compared to the other genes between *S. tristyla* and the kiwifruit, indicating possible positive selection. The Ka/Ks values of most PCGs were less than 1, such as *atp9*, *ccmB*, *ccmC*, *cox3*, *nad6* and *rps12*, indicating that most PCGs were under purification selection (Figure 6). These results suggested that most PCGs may be highly conservative in the evolutionary process of Actinidiaceae.

In order to further analyze the phylogenetic position of Actinidiaceae, 23 plant mitogenomes from GenBank were downloaded for phylogenetic tree construction based on 20 PCGs. Phylogenetic analysis showed that 23 plant mitogenomes were divided into 6 categories (Figure 7). We selected *V. vinifera* and *N. nucifera* as outgroups. The phylogenetic tree strongly demonstrated that five kiwifruit species (*A. chinensis* (2×), *A. chinensis* (4×), *A. chinensis* var deliciosa (6×), *A. eriantha* and *A. arguta*) clustered into one clade with a 100% bootstrap value (Figure 7). It also revealed that *S. tristyla* was closely related to five kiwifruit species (Figure 7).

## 4. Discussion

So far, 25 *Actinidia* genus chloroplast genomes have been reported [32], and our group has comprehensively analyzed them. Unlike conserved genome structures and small *Actinidia* chloroplast genomes, *Actinidia* mitogenomes generally have multiple different sizes and structural variations [55]. This makes *Actinidia* mitogenome research relatively challenging. Here, complete mitogenomes of five Actinidiaceae plants were sequenced and assembled. Two subgenomic circles were found in *A. chinensis* (2×–4×) and *S. tristyla.* Similar results have been reported in *C. sinensis* var. assamica and *C. assamica* [16,56]. The GC content of five Actinidia mitogenomes is about 46%, similar to most other mitogenomes [57]. Among the observed size variations, the genome size in *A. chinensis* var deliciosa (6×) was about twice that of the *S. tristyla* (482 kb) and the closely related species *Vaccinium macrocarpon* (459 kb) [58]. The genome size of the hexaploid *A. chinensis* var deliciosa was nearly 20 kb larger than that of diploid and tetraploid *A. chinensis*, which is probably the result of a gradual increase in sequence duplication and intracellular transfer of the plastid or nuclear genome or horizontal transfer of mitochondrial DNA during evolution [59,60].

Repeat sequences widely exist in plant mitogenomes, including tandem repeats and SSRs [61]. Positive correlations between genome size and repeat sequences were identified in 38 Rosaceae mitogenomes [62]. As shown in Figure 2A, tandem repeats from 10 to 30 bp were the most abundant for the Actinidiaceae mitogenomes, with similar results in *Diospyros oleifera* [63]. Guo et al. [64] have reported that SSRs played a pivotal role in intermolecular recombination during evolution. The number of SSRs in the *A. chinensis* mitogenome (18.24%) was higher than that of *S. tristyla* and *A. eriantha* (Figure 2B), which may cause the *A. chinensis* mitogenome size to be larger than *S. tristyla* and *A. eriantha.* It is consistent with the findings of previous studies [65,66]. Dimer repeats were the most abundant SSR type (about 48%) in the *Actinidia* mitogenome (Figure 2C), which is commonly found in *Suaeda glauca* [67]. Gene transfer from the chloroplast to the mitogenome frequently occurs during long-term plant evolution [68]. A total of 9–55 kb of plastid-derived sequences was observed, which occupied 3–6% of the Actinidiaceae mitogenomes (Figure 3). Similar results have been reported by Adams et al. [69]. Some plastid-derived protein-coding genes (cp-derived PCGs), such as *rpoC1*, *ndhB*, *rps7*, *rps19*, and *rpl23*, were identified in the *A. chinensis* (2×–4×) mitogenomes, which is commonly found in angiosperms [70]. In addition, we also found that *psbJ*, *petL* and *petG* cp-derived genes only exist in hexaploid *A. chinensis* var deliciosa (Appendix A), suggesting that special evolutionary events may have occurred during genome evolution.

Mitochondrial gene expression may be affected by RNA editing [71]. The number of RNA editing sites varies in different species [72]. Previous studies identified approximately 491 RNA editing sites within 34 genes in rice [68] and 486 RNA editing sites within 31 genes in *Primula vulgaris* [73]. In Actinidiaceae, the number of RNA editing sites in most PCGs was extremely conserved in Actinidiaceae, similar to other plant studies [74]. Interestingly, we also observed that the number of NAD1 gene RNA editing sites increased with ploidy in *A. chinensis* (2×–4×) (Figure 4). Whether the number of RNA editing sites is positively correlated with the ploidy of the kiwifruit requires further research. Relative synonymous codon usage (RSCU) refers to the relative probability of a specific codon between the synonymous codons that encode the corresponding amino acid [75]. The RSCU value showed that the codon usage pattern in the Actinidiaceae plants’ mitogenomes shared a similar RSCU style (Figure 5A–E), which was commonly found in higher plant mitogenomes [76].

Calculating the Ka/Ks ratio plays a vital role in understanding the dynamics of molecular evolution [77]. As shown in Figure 6, the Ka/Ks ratios of most PCGs were less than 1 (Figure 6), suggesting that these genes were highly conserved and had undergone neutral and negative selections in Actinidiaceae. These results were also supported in the report on Lamiales [78]. However, among the five Actinidiaceae plants, sdh3 with dN/dS values greater than 1.0 was found between the *S. tristyla* and kiwifruit mitogenomes, indicating that this gene may have suffered from positive selection during the evolution in Actinidiaceae. The phylogenetic trees in this study showed a close relationship between the Actinidiaceae and other Ericaceae plants (Figure 7), as Wang et al. [56] proposed. Notably, our analyses also demonstrated that the Actinidiaceae is monophyly, with the sampled five *Actinidia* taxa clustering in a clade as a sister to *S. tristyla* (Figure 7), in agreement with the result of Wang et al. [32]. Moreover, *A. chinensis* (2×–4×) was closely related to *A. chinensis* var deliciosa (6×) (Figure 7), which was consistent with the results of a previous study [79].

## 5. Conclusions

The large size variation in Actinidiaceae mitogenomes appeared due to increasing sequence duplication and intracellular transfer of the plastid. The number of RNA editing sites and codon usage in most PCGs of five Actinidiaceae plants’ mitogenomes were highly conserved. Most of the coding genes had undergone negative selection, indicating the conservation of mt genes during evolution. We found that sdh3 may have suffered from positive selection during the evolution in Actinidiaceae. Kiwifruit species showed high similarities and were highly similar to *S. tristyla* and *A. chinensis* (2×–4×) was closely related to *A. chinensis* var deliciosa (6×). This study provides important mitochondrial genome resources for the Actinidiaceae species and has deepened our understanding of organelle genome evolution in flowering plants.

## Figures and Tables

**Figure 1 genes-13-01827-f001:**
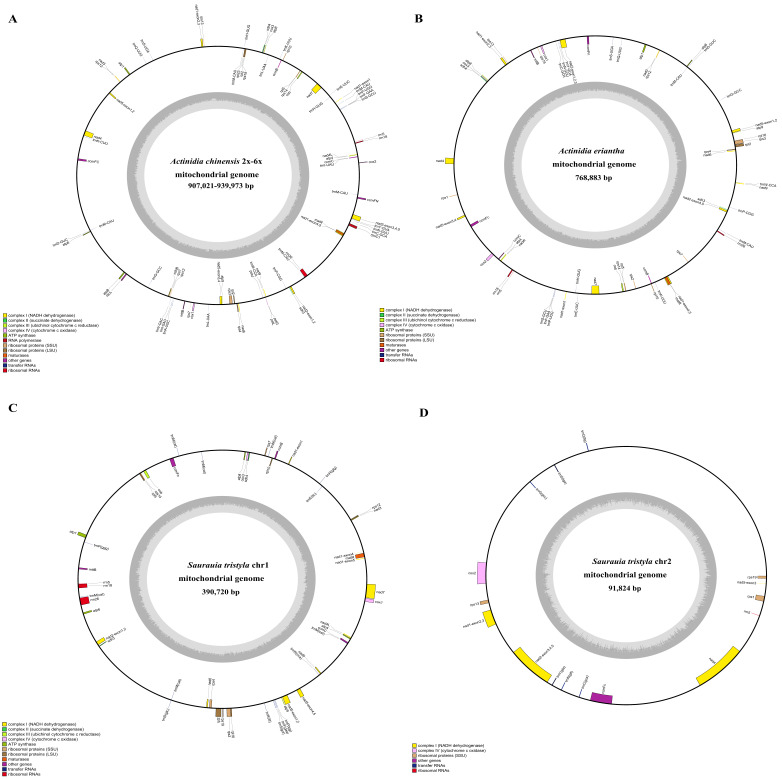
(**A**–**D**) Gene map of four kiwifruit species (*A. chinensis* (2×–6×), *A. eriantha*) and *S. tristyla,* representing the mitogenome structure. Genes drawn outside the circle are transcribed clockwise, and those inside are counterclockwise. Genes that belong to different functional groups are color coded. The darker grey in the inner circle indicates the GC content of the mitogenome.

**Figure 2 genes-13-01827-f002:**
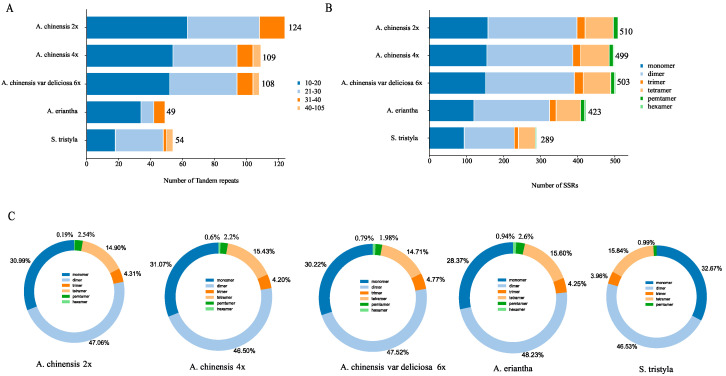
The tandem and SSR repeats in five Actiniaceae mitogenomes. (**A**) The number of tandem repeats. (**B**) The number of SSRs. (**C**) Pie chart for SSR distribution. The colors represent different types of SSRs.

**Figure 3 genes-13-01827-f003:**
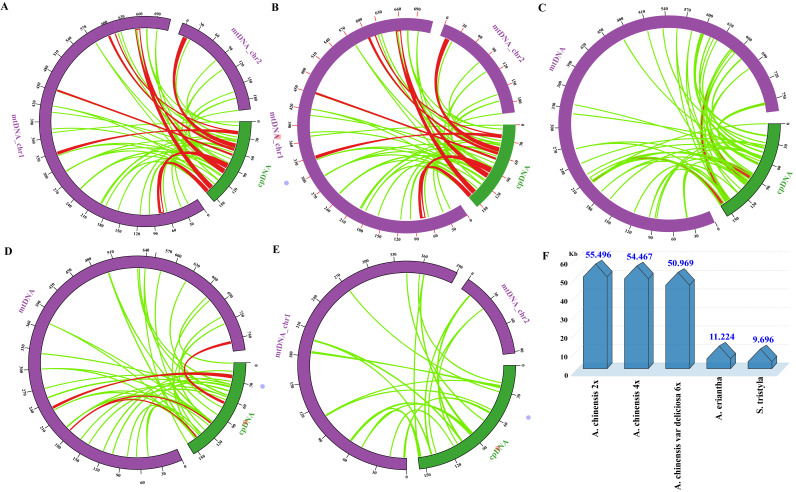
(**A**–**E**) Shared genome regions of each species between chloroplasts and mitochondria of *A. chinensis* (2×), *A. chinensis* (4×), *A. chinensis* var deliciosa (6×), *A. eriantha* and *S. tristyla*, respectively. The green circular segment represents the mitogenome, and the purple circular segment represents the chloroplast genome. (**F**) Shared sequence length of each species between chloroplasts and mitochondria genomes.

**Figure 4 genes-13-01827-f004:**
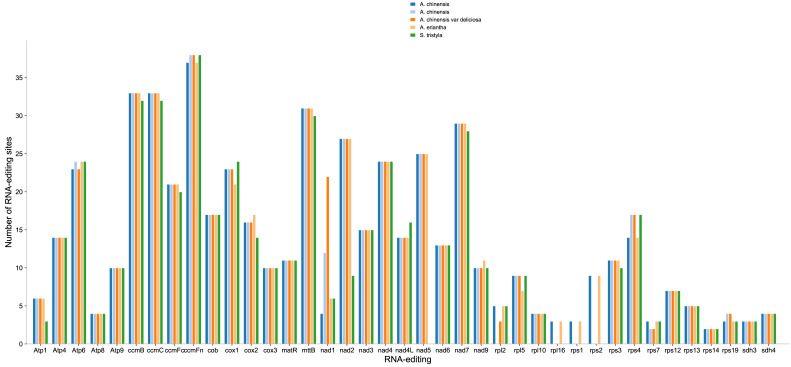
The distribution of RNA editing sites in mitogenome protein-coding genes.

**Figure 5 genes-13-01827-f005:**
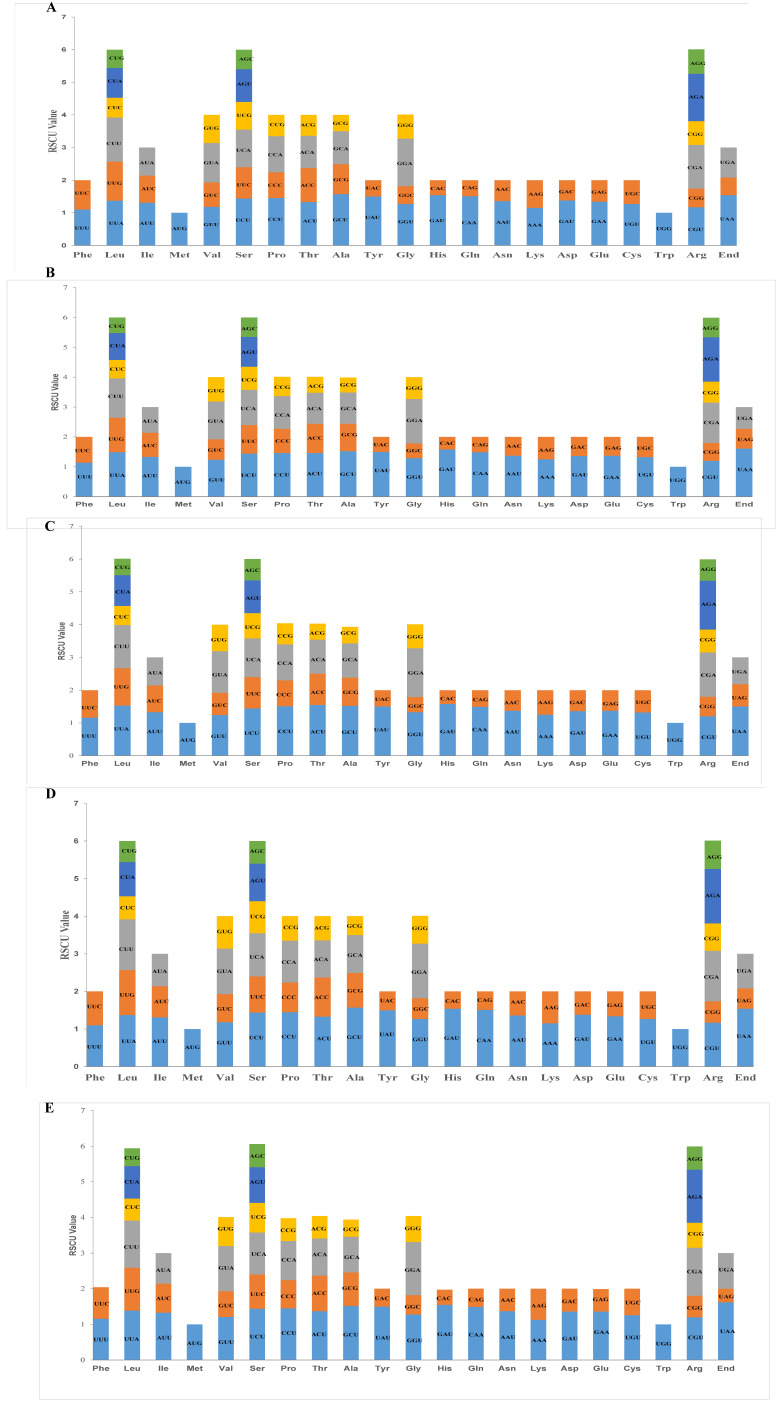
Relative synonymous codon usage (RSCU) in mitochondrial protein-coding genes of five *Actinidiaceae* mitogenomes. The y-axis represents the value for RSCU. (**A**) The RSCU value of *A. chinensis* (2×). (**B**) The RSCU value of *A. chinensis* (4×). (**C**) The RSCU value of *A. chinensis* var deliciosa (6×). (**D**) The RSCU value of *A. eriantha*. (**E**) The RSCU value of *S. tristyla*.

**Figure 6 genes-13-01827-f006:**
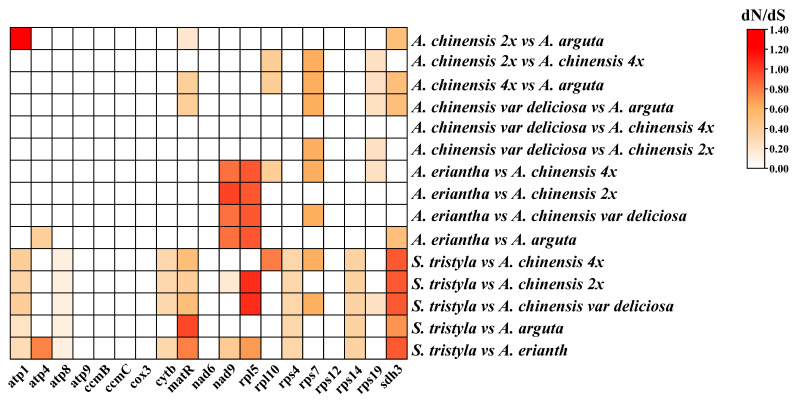
The pairwise Ka/Ks ratios among each mitochondrial gene in the six Actinidiaceae family species.

**Figure 7 genes-13-01827-f007:**
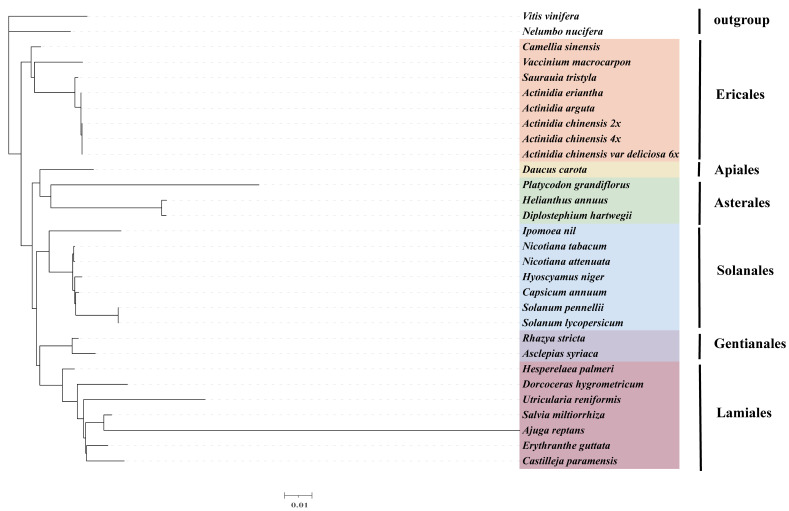
Maximum likelihood phylogenetic tree analysis of Actinidiaceae mitogenomes based on 20 PCGs of 23 plant mitogenomes with *V. vinifera* and *N. nucifera* as outgroups.

## Data Availability

The accession number generated for this study can be found in Appendix A.

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
