# Peer review of "A Comparative Genomics Approach for Analysis of Complete Mitogenomes of Five Actinidiaceae Plants"

_genes, 2022, doi:10.3390/genes13101827_

Round 1

Reviewer 1 Report

The manuscript "Complete Mitochondrial Genome of Five Actinidiaceae Plants and Identification of the Actinidia Chinensis Ploidy Based on Mitochondrial DNA Marker" is a nicely undertaken attempt to explore mitochondrial genome complexity in Kiwifruit and related species. It will help workers for identification as well as Phylogenetic studies in Kiwifruit. Although the manuscript is well structured, there are segments to improve like Discussion, conclusion and the English language, grammatical punctuations etc. The title of the manuscript is appropriate, but in the "Actinidia chinensis" species name must be started with a small letter. The abstract section is well written except for some sentences that start with "We" and Our. Some of the key findings may also be included in the abstract. As the Kiwifruit is neither in the title nor in key words, it can be included in place of comparative analysis.

In line no. 40: It has been written "mitochondrial genome is more difficult and complex than that of other organelles." As the analysis process is in-silico, it can be written that "the interpretation and functional annotation of the mitochondrial genome is complex in comparison to other organelles". The term "more difficult" is inappropriate.

line no. 55 to 57: Rephrase the sentences from as it is difficult to understand.
Line No. 58: there is a spelling mistake "iessential" also it is not clear which feature authors want to say.

Line No. 81 to 82: "Abiotic and biotic stresses seriously affect the yield and quality of kiwifruit, especially Pseudomonas syringae pv. actinidiae (Psa) tolerance in A. chinensis [24]." is contradictory, rewrite it.

Line No. 83: "Once Psa occurs, removing all kiwifruit trees is necessary" it can be written as "After incidence of Psa, there is no remedy available to control it till date except destroying the tree to prevent the spread of disease."

Line No. 86-87 the word "resilience" will be more suitable than "resistance" covering both the abiotic as well as biotic stresses.

Line No. 88-89 "mitochondria genome sequences of kiwifruit are sequenced" it should be "mitochondrial genome of kiwifruit are sequenced".

Line No. 96 "were assembled in this study" is it assembled only? or "sequenced and assembled in this study".

Line No. 99-102 "....structures of four kiwifruit species and S. tristyla, genomic repeats, RNA editing sites...." it should be "structures of four kiwifruit species and S. tristyla to elucidate/unveil genomic repeats, RNA editing sites...."

Line No. 107-108 "Fresh leaves were wrapped in foil, frozen in flash nitrogen, and stored at −80°C for subsequent use." It should be " Fresh leaves were wrapped in aluminum foil, flash frozen in liquid nitrogen, and stored at −80°C for subsequent use."

Line No. 114-116 "...were conducted following a polished..." can be better written as "...were conducted after polishing...

Line No. 117 "assembled" should be "assemble".

Line No. 118 "Finally, the assembled PacBio sequences were checked for overlaps and 118 were connected." may be written as "In the last step the assembled PacBio sequences were checked for overlaps and joined."

Line No. 126 "Simple Repeated Sequence" must be " Simple Sequence Repeats".
Line No. 153 The word "Additionally" may be omitted or replaced with "Further".

Line No. 154 In the heading "Ploidy" should be "Ploids".

Line No. 156 The "Primers" should be defined as "Nad1 Intron Insertion specific Primers".

Line No. 158 The "25 μl mixture" should be "25 μl reaction mixture"

Line No. 162 As the "Sanger sequencing" has been performed, the sequence reads can be supplemented as supplementary data or deposited in NCBI and sequence ids may be provided.
Line No. 187 "belonging to between 10-20" should be "comprising between 10-20bp".

Line No. 194 "Etramers" should be "Tetramers".

Line No. 195 "were less numerous" may be written as "were less frequent".

Line No. 196 "more" may be replaced with "higher number of" or "much number of".

Line No. 196 "S. tristyla had the least SSRs" should be "S. tristyla had the least number of SSRs".
Line No. 201 Figure caption "The Tandem and SSR repeat in five Actiniaceae Mitochondrial genome." It should be "The tandem and SSR repeats in five Actiniaceae mitochondrial genomes."
Line No. 202 Figure caption 2 (C) "The percentages of SSRs were also provided on the 202 pie chart." should be "Pie chart for SSRs distribution."

 Line No. 207 "transfer" should be "transferred".

Line No. 216 "Shared regions" should be "Shared genome regions".

Line No. 225 ".....were the most RNA-editing sites in five Actinaceae species." should be ".....were having the most of RNA-editing sites in five Actinaceae sp."

 Line No. 273 "Development of A Mitochondrial NAD1 Intron Marker....." Should be "Detection of Ploidy levels of A. chinensis based on novel NAD1 Intron Markers".

Line No. 278-279 Re phrase is as it is difficult to understand.

 Line No. 289 In "Figure 8 caption there should be two segments (A) and (B). "B Alignment...." is confusing. it should be "(B) Alignment...."

 Line No. 293 "......and our group has performed a comprehensive analysis of them." Should be rephrased.

 Line No. 310 "....includes tandem,..." should be "....includes tandem repeats,...".

 The Authors should have proper use of Bases and base pairs. At various places in the manuscript "Bases" have been written in place of "bp".

Authors should use any one among these terms rather than using all these "mt genome"; "mitochondrial genome"; "mitogenomes".

The Discussion and Conclusions sections should focus on salient findings and interpretations rather than repeated description of the results.

Author Response

Reviewer, Genes

October 29, 2022

Dear Reviewer

Thank you so much for critiquing our submission. We reviewed the concerns from the reviewers and have improved our manuscript. we have incorporated the suggested changes and would like to resubmit our manuscript. We have incorporated the changes suggested by the reviewers and have answered their questions to the best of our ability. Please see our point-to-point responses below.

Reviewer 1:

Comments and Suggestions for Authors

The manuscript "Complete Mitochondrial Genome of Five Actinidiaceae Plants and Identification of the Actinidia Chinensis Ploidy Based on Mitochondrial DNA Marker" is a nicely undertaken attempt to explore mitochondrial genome complexity in Kiwifruit and related species. It will help workers for identification as well as Phylogenetic studies in Kiwifruit.

comments

Although the manuscript is well structured, there are segments to improve like Discussion, conclusion and the English language, grammatical punctuations etc.

A: First of all thank you for your appreciation. We have done our best to revise discussion, conclusion and the English language, grammatical punctuations etc to improve the quality of articles.

The title of the manuscript is appropriate, but in the "Actinidia chinensis" species name must be started with a small letter.

A: We have changed the title of the paper to A Comparative Genomics Approach for Analysis of Complete Mitochondrial Genomes of Five Actinidiaceae Plants”.

The abstract section is well written except for some sentences that start with "We" and Our.

A: We have rewritten the sentences with "We" and "Our" in abstract section.

As the Kiwifruit is neither in the title nor in key words, it can be included in place of comparative analysis.

A: we have added “the Kiwifruit” in place of comparative analysis in line 22-23.

In line no. 40: It has been written "mitochondrial genome is more difficult and complex than that of other organelles." As the analysis process is in-silico, it can be written that "the interpretation and functional annotation of the mitochondrial genome is complex in comparison to other organelles". The term "more difficult" is inappropriate.

A: We have changed "mitochondrial genome is more difficult and complex than that of other organelles." to "the interpretation and functional annotation of the mitochondrial genome is complex in comparison to other organelles" in line 46-48.

Specific comments

line no. 55 to 57: Rephrase the sentences from as it is difficult to understand.

A: We have rephrase the sentences in line 70-71.

Line No. 58: there is a spelling mistake "iessential" also it is not clear which feature authors want to say.

A: We have changed "iessential" to "essential" in line 72.

Line No. 81 to 82: "Abiotic and biotic stresses seriously affect the yield and quality of kiwifruit, especially Pseudomonas syringae pv. actinidiae (Psa) tolerance in A. chinensis [24]." is contradictory, rewrite it.

A: We have rewrite this senence in line 96-98.

Line No. 83: "Once Psa occurs, removing all kiwifruit trees is necessary" it can be written as "After incidence of Psa, there is no remedy available to control it till date except destroying the tree to prevent the spread of disease."

A: We have changed " Once Psa occurs, removing all kiwifruit trees is necessary " to " After incidence of Psa, there is no remedy available to control it till date except destroying the tree to prevent the spread of disease." in line 99-101.

Line No. 86-87 the word "resilience" will be more suitable than "resistance" covering both the abiotic as well as biotic stresses.

A: We have changed " resistance resilience" to " resilience " in line 105.

Line No. 88-89 "mitochondria genome sequences of kiwifruit are sequenced" it should be "mitochondrial genome of kiwifruit are sequenced".

A: We have changed "mitochondria genome sequences of kiwifruit are sequenced" to " mitogenomes of kiwifruit are sequenced" in line 106.

Line No. 96 "were assembled in this study" is it assembled only? or "sequenced and assembled in this study".

A: We have changed "assembled " to " sequenced and assembled " in line 115.

Line No. 99-102 "....structures of four kiwifruit species and S. tristyla, genomic repeats, RNA editing sites...." it should be "structures of four kiwifruit species and S. tristyla to elucidate/unveil genomic repeats, RNA editing sites...."

A: We have changed "....structures of four kiwifruit species and S. tristyla, genomic repeats, RNA editing sites...." to " structures of four kiwifruit species and S. tristyla to elucidate/unveil genomic repeats, RNA editing sites.... " in line 121-122.

Line No. 107-108 "Fresh leaves were wrapped in foil, frozen in flash nitrogen, and stored at −80°C for subsequent use." It should be " Fresh leaves were wrapped in aluminum foil, flash frozen in liquid nitrogen, and stored at −80°C for subsequent use."

A: We have changed " Fresh leaves were wrapped in foil, frozen in flash nitrogen, and stored at −80°C for subsequent use." to " Fresh leaves were wrapped in aluminum foil, flash frozen in liquid nitrogen, and stored at −80°C for subsequent use." in line 131-132.

Line No. 114-116 "...were conducted following a polished..." can be better written as "...were conducted after polishing... "

A: We have changed ".....were conducted following a polished...." to " were conducted after polishing " in line 140.

Line No. 117 "assembled" should be "assemble".

A: We have changed " assembled " to " assemble " in line 143.

Line No. 118 "Finally, the assembled PacBio sequences were checked for overlaps and 118 were connected." may be written as "In the last step the assembled PacBio sequences were checked for overlaps and joined."

A: We have changed "Finally, the assembled PacBio sequences were checked for overlaps and 118 were connected." to " In the last step the assembled PacBio sequences were checked for overlaps and joined." in line 143-144.

Line No. 153 The word "Additionally" may be omitted or replaced with "Further".

A: We have changed " Additionally " to " Further " in line 180.

Line No. 154 In the heading "Ploidy" should be "Ploids".

A: We have deleted "Ploidy".

Line No. 156 The "Primers" should be defined as "Nad1 Intron Insertion specific Primers".

A: We have deleted " Primers ".

Line No. 158 The "25 μl mixture" should be "25 μl reaction mixture"

A: We have deleted " 25 μl mixture ".

Line No. 162 As the "Sanger sequencing" has been performed, the sequence reads can be supplemented as supplementary data or deposited in NCBI and sequence ids may be provided.

A: We have deleted " Sanger sequencing ".

Line No. 187 "belonging to between 10-20" should be "comprising between 10-20bp".

A: We have changed " belonging to between 10-20" to " comprising between 10-20bp " in line 217.

Line No. 194 "Etramers" should be "Tetramers".

A: We have changed " Etramers " to " Tetramers " in line 224.

Line No. 195 "were less numerous" may be written as "were less frequent".

A: We have changed " were less numerous " to " were less frequent " in line 224.

Line No. 196 "more" may be replaced with "higher number of" or "much number of".

A: We have changed " higher number of " to " much number of  " in line 226.

Line No. 196 "S. tristyla had the least SSRs" should be "S. tristyla had the least number of SSRs".

A: We have changed " S. tristyla had the least SSRs " to " S. tristyla had the least number of SSRs " in line 226-227.

Line No. 201 Figure caption "The Tandem and SSR repeat in five Actiniaceae Mitochondrial genome." It should be "The tandem and SSR repeats in five Actiniaceae mitochondrial genomes."

A: We have changed " The Tandem and SSR repeat in five Actiniaceae Mitochondrial genome." to " The tandem and SSR repeats in five Actiniaceae mitogenomes. " in line 232.

Line No. 202 Figure caption 2 (C) "The percentages of SSRs were also provided on the 202 pie chart." should be "Pie chart for SSRs distribution."

A: We have changed " The percentages of SSRs were also provided on the 202 pie chart." to " Pie chart for SSRs distribution." in line 234.

 Line No. 207 "transfer" should be "transferred".

A: We have changed " transfer " to " transferred " in line 240.

Line No. 216 "Shared regions" should be "Shared genome regions".

A: We have changed " Shared regions " to " Shared genome regions " in line 249.

Line No. 225 ".....were the most RNA-editing sites in five Actinaceae species." should be ".....were having the most of RNA-editing sites in five Actinaceae sp."

A: We have changed " were the most RNA-editing sites in five Actinaceae species. " to " were having the most of RNA-editing sites in five Actinaceae sp " in line 258-259.

Line No. 273 "Development of A Mitochondrial NAD1 Intron Marker....." Should be "Detection of Ploidy levels of A. chinensis based on novel NAD1 Intron Markers".

A: We have deleted " Development of A Mitochondrial NAD1 Intron Marker " in line 308.

Line No. 278-279 Re phrase is as it is difficult to understand.

A: We have Rephrase Line No. 278-279 in line 328-329.

 Line No. 289 In "Figure 8 caption there should be two segments (A) and (B). "B Alignment...." is confusing. it should be "(B) Alignment...."

A: We recently added the ploidy detection of different ploidy kiwifruit using NAD1 marker, found that some ploidy was incorrectly identified, so we deleted section 3.5.

 Line No. 293 "......and our group has performed a comprehensive analysis of them." Should be rephrased.

A: We have rephrased "......and our group has performed a comprehensive analysis of them."  to " and our group has comprehensively analysed them." in line 328-329.

 Line No. 310 "....includes tandem,..." should be "....includes tandem repeats,...".

A: We have changed " ....includes tandem,.." to " ....includes tandem repeats,... " in line .

 The Authors should have proper use of Bases and base pairs. At various places in the manuscript "Bases" have been written in place of "bp".

A: We have changed " bp " to " Bases " At various places.

Authors should use any one among these terms rather than using all these "mt genome"; "mitochondrial genome"; "mitogenomes".

A:  We have only used "mitogenomes" in our whole article.

The Discussion and Conclusions sections should focus on salient findings and interpretations rather than repeated description of the results.

A:  We have deleted repeated description of the results and added some relevant literatures focus on salient findings in Discussion and Conclusions.

Best regards,

Jun Yang

Reviewer 2 Report

The manuscript shows interesting data related to “Complete Mitochondrial Genome of Five Actinidiaceae Plants and Identification of the Actinidia Chinensis Ploidy Based on Mitochondrial DNA Marker”.  Although I have seen merit in the study and the data may be of interest for a general reader. I have revised the entire manuscript carefully and prepared below a detailed list of corrections and suggestions that will certainly improve the quality and make the text more evident and scientific.

1.      The technical information presented in the paper was interesting and worth reading.

2.      The abstract was informative and it was reflecting the body of the paper.

3.      Acronyms must be written in full form where first time used. See line 44-45 “Differences in mt genomes size can be attributed to repetitive sequences and foreign DNA derived from other organisms during evolution”.

4.      Add a research hypothesis in the last paragraph of introduction section.

5.      The results section is well written and it is easy to understand.

6.     Add some more relevant literature to the discussion section and discuss your finding in more scientific language.

7.      Improve the quality of figures.

8.      Rewrite the conclusion section in more technical way.

9.      The reference section was informative. However, incorporate some latest references and arrange them according to the journal pattern.

My Final Decision

Minor Revision

Author Response

Reviewer, Genes

October 29, 2022

Dear Reviewer

Thank you so much for critiquing our submission. We reviewed the concerns from the reviewers and have improved our manuscript. we have incorporated the suggested changes and would like to resubmit our manuscript. We have incorporated the changes suggested by the reviewers and have answered their questions to the best of our ability. Please see our point-to-point responses below.

Reviewer 2:

Comments and Suggestions for Authors

The manuscript shows interesting data related to “Complete Mitochondrial Genome of Five Actinidiaceae Plants and Identification of the Actinidia Chinensis Ploidy Based on Mitochondrial DNA Marker”.  Although I have seen merit in the study and the data may be of interest for a general reader. I have revised the entire manuscript carefully and prepared below a detailed list of corrections and suggestions that will certainly improve the quality and make the text more evident and scientific.

comments

  1. The technical information presented in the paper was interesting and worth reading.

A: We appreciate it very much for your approval.

  1. The abstract was informative and it was reflecting the body of the paper.

A: We appreciate that the abstract meet with your approval.

  1. Acronyms must be written in full form where first time used. See line 44-45 “Differences in mt genomes size can be attributed to repetitive sequences and foreign DNA derived from other organisms during evolution”.

A: We have changed " mt genomes" to " mitogenomes " in line 53.

  1. Add a research hypothesis in the last paragraph of introduction section.

A: we have added hypothesis in the last paragraph of introduction section in line 120-121.

  1. The results section is well written and it is easy to understand.

A: Thanks again for your approval.

  1. Add some more relevant literature to the discussion section and discuss your finding in more scientific language.

A: We have add some more relevant literature in the discussion section in more scientific language.

  1. Improve the quality of figures.

A: We provide PDF and images above 300dpi in the attachment.

  1. Rewrite the conclusion section in more technical way.

A: We have done our best to rewrite the conclusion section in more technical way.

Best regards,

Jun Yang
